# Use and Preferences of Health Apps among Women and Healthcare Professionals Regarding GDM Postpartum Care Related to Diet, Physical Activity, and Weight Management: A Cross-Sectional Survey

**DOI:** 10.3390/nu15153304

**Published:** 2023-07-25

**Authors:** Anna Roesler, Kaley Butten, Cobi Calyx, Elizabeth Holmes-Truscott, Pennie Taylor

**Affiliations:** 1The Australian e-Health Research Centre, CSIRO, Herston, QLD 4006, Australia; kaley.butten@csiro.au (K.B.); pennie.taylor@csiro.au (P.T.); 2School of Humanities and Languages, UNSW Sydney, Sydney, NSW 1466, Australia; 3School of Psychology, Institute for Health Transformation, Deakin University, 1 Gheringhap St., Geelong, VIC 3220, Australia; etruscott@acbrd.org.au; 4The Australian Centre for Behavioural Research in Diabetes, Diabetes Victoria, 15-31 Pelham Street, Carlton, VIC 3053, Australia

**Keywords:** mHealth, gestational diabetes, health professionals, obesity, apps, postpartum

## Abstract

Gestational Diabetes Mellitus (GDM) is a common medical complication of pregnancy, which is associated with increased risk of future diabetes. mHealth (mobile health, in this paper applications abbreviated to apps) can facilitate health modifications to decrease future risks. This study aims to understand mHealth app use and preferences among women with past GDM and healthcare professionals (HCP) in Australia. An explorative cross-sectional online survey was disseminated via social media, a national diabetes registry, and professional networks. Descriptive analyses were conducted on valid responses (women with prior GDM: *n* = 1475; HCP: *n* = 75). One third (33%) of women with prior GDM have used health apps, and a further 80% of non-app users were open to using a health app if recommended by their HCP. Over half (53%) of HCPs supported health information delivery via mHealth, although only 14% had recommended a health app to women post-GDM, and lack of knowledge about mHealth apps was common. Health app users reported that they preferred tracking features, while non-users desired credible health and dietary information and plans. Expanding mHealth app use could facilitate healthy behaviours, but endorsement by HCPs is important to women and is still currently lacking.

## 1. Introduction

Gestational diabetes mellitus (GDM) is a common medical complication of pregnancy, affecting 13% of pregnancies globally [1], and slightly more in Australia, at 17% [2]. Women diagnosed with GDM have an approximate 40% risk of a recurrence of GDM in a subsequent pregnancy (The Royal Australian College of General Practitioners 2020) and are seven times more likely to develop type 2 diabetes in the future (T2D) [3]. Risk factors for GDM include either personal or family history of diabetes, living with obesity or overweight, physical inactivity, age, polycystic ovary syndrome, use of some medication, ethnicity [4], and dietary factors [5]. Of these risk factors, meeting dietary and physical activity recommendations, along with obtaining and maintaining a healthy weight, are recommended for managing diabetes [6] and reducing the risk of GDM in future pregnancies and T2D [6,7].

Diet and physical activity recommendations extend to the postpartum period and are universally endorsed [8] as the first line treatment for preventing or delaying progression towards T2D. However, the delivery and adoption of these recommendations postpartum remains a challenge. Numerous barriers have been reported in the literature relating to various individual and system-level factors (including but not limited to: insufficient time and resources due to the competing demands of motherhood, limited social supports, shifting focus from maternal to child health, lack of care-coordination, lack of culturally responsive health care, and limited communication opportunities between hospital, primary care, and patients).

Mobile health (mHealth) has been identified as a potential solution to help facilitate diet and physical activity recommendations in the postpartum period, without the potentially onerous time and travel requirements that are generally associated with in-person interventions [9,10,11,12,13]. There is an abundance of publicly available digital health applications (apps), with more than 90,000 new health apps available in 2020 (IQVIA Institute, Parsippany, NJ, USA, 2021). However, evidence for the frequency of health app usage among women post-GDM remains unclear [14]. Studies in the general population have reported limited to no engagement [15], acceptability [16], or effectiveness [17] with health apps, while some studies in women—including during GDM—have demonstrated that health apps improved treatment uptake, self-awareness, and self-management [18,19]. Indeed, Lim et al.’s [14] review of qualitative studies of digital health interventions for postpartum women concluded that a digital approach was well accepted by women and should be considered in developing postpartum behaviour change strategies going forward.

While evidence of mHealth’s effectiveness post-GDM is as yet unconvincing, health apps and online programs for women post GDM are nonetheless being developed, evaluated, and, in some instances, rolled out [20,21,22,23,24,25,26,27,28,29,30,31]. Apps and online programs designed (or adapted) for women with a prior history of GDM vary in their content, design, availability, and delivery method. For example, in the UK, Baby Steps is a structured group education program with an accompanying online program designed to promote physical activity and other health behaviours among women with prior experience of GDM [23,32,33]. Features of the accompanying online program include an interactive education component and a variety of resources using different formats (such as video animations, expert videos, interactive activities, and quizzes) which supplement the messages delivered during the group sessions. In Australia, only the online program component of Baby Steps has been adopted and rolled out via the National Diabetes Services Scheme (NDSS), an Australian Government Initiative administered by Diabetes Australia. However, evidence on the adoption, acceptability, and effectiveness of such online mHealth tools for women post-GDM is limited. Furthermore, it remains unknown what mHealth tools women with prior experience of GDM are currently accessing to support their health behaviour change, not to mention what content and features such women would want.

Most health apps, and online programs, are standalone—publicly available, outside of the health system, and unregulated. These mHealth tools provide health information and behaviour change advice that is potentially unrelated to evidence-based clinical guidelines. Nonetheless, women are using these health apps [34]. Australian healthcare professionals (HCPs) may not be comfortable recommending apps to patients, with one study indicating that this is due to a lack of GP knowledge regarding which apps are effective or trustworthy [35]. This matches the findings of a UK study which reported that mHealth resources were rarely recommended by HCPs [36]. Thus, in addition to understanding women’s experiences of mHealth app use, more research on current app recommendations and the preferences of HCPs is needed.

This study therefore aims to explore postpartum health information and support needs, along with current use of and preferences for health apps among women with prior GDM. This is important as we are unaware of the current usage and preferences of health apps, information which is required to understand how to better use these technologies to support women’s health. In addition, we explore the comparative views of health professionals who work with women with GDM, more specifically their views on postpartum support needs and the use of apps.

## 2. Materials and Methods

This explorative study took the form of a cross-sectional online survey, led by a multi-disciplinary team with expertise in GDM, dietetics, public health, health promotion, and health psychology, including researchers with lived experience of GDM (*n* = 2). Ethical approval was granted by CSIRO Health and Medical Human Research Ethics Committee (ID 2022_061_LR). Data collected from this study have not been deposited in a publicly available database, due to associated license agreements and commercial viability.

### 2.1. Study Participants

#### 2.1.1. Women with Prior GDM

Women with prior GDM were recruited (November 2022–March 2023) via two main avenues. First, the survey was publicly shared via online social media (Facebook, Twitter, and LinkedIn), websites, and e-newsletter via the researchers and affiliated organizations, including paid Facebook advertising targeting women aged 18–45. Second, a direct email invitation was distributed by the NDSS to a random sample of 40,000 registrants with prior experience of GDM who had consented to receive information about research opportunities. The NDSS provides subsidized access to diabetes programs and services in Australia, with >40,000 women with GDM newly registered annually over the past five years [37]. All survey promotions included a link to the survey.

Participants were only eligible to take part in the survey if (1) they had experienced GDM within the last five years and given birth for that pregnancy, (2) had received GDM care within Australia, (3) had not been diagnosed with type 1 diabetes or T2D prior to their pregnancy with GDM, and (4) were aged 18 years or older. Survey completion was incentivized with the chance of winning a $25AUD gift voucher.

#### 2.1.2. Healthcare Professionals

In a separate survey, HCPs were recruited via online social media (Facebook, Twitter, and LinkedIn), website promotions, as well as direct email to Australian national and state-based diabetes organizations and professional associations inviting them to share the survey link widely. The HCP survey was open between February–March 2023.

Participants were eligible if they had worked in Australia in the last five years to provide diabetes care to women who had been identified as having risk factors for GDM, had GDM, or experienced GDM postpartum.

#### 2.1.3. Sample Size

The minimum sample size for the women’s survey was estimated by using the formula of the population proportion estimation. The criterion of maximum variability was applied, with a 95% confidence interval and a 5% margin of error. A minimum sample size of 384 women was required. However, as we are completing a descriptive analysis of women within five years of experiencing GDM, we calculated what number of women would give us 1% of this population. The number of women with GDM in Australia was taken from the National Hospital Morbidity Database (15 July 2020 AIHW), indicating that 2155 women would represent 1% of the population. We therefore set the parameters of a representative sample between 384–2155. The HCP’s survey sample size was determined based on ensuring there was representation from each state.

### 2.2. Procedure

Potential participants were directed to the relevant survey (on REDCap(r) (Research Electronic Data Capture 13.1.29) an electronic data capture tool hosted at CSIRO) which included plain language study information, sought informed consent, and screened for eligibility. Ineligible participants were automatically screened out while eligible participants were directed to the survey proper. At survey completion participants could “opt-in” to the participant prize draw by providing their contact details (stored independently of survey responses). Survey data was automatically saved, retaining confidential responses of participants who dropped off. The median (IQR) survey duration was 10 min (5–18 min) for women with GDM and 8 min (5–16 min) for HCPs, respectively.

### 2.3. Survey Measures

Survey measures included study-specific closed- (multiple choice, Likert) and open-ended (i.e., free text) questions designed by the research team, with input from women with GDM, HCPs, and researchers with expertise in GDM. Six women and five HCPs that met the survey eligibility criteria pre-tested the survey tools and provided written feedback that was then used to refine the survey. The survey was further refined in an iterative process as insights were gained from reviewing participant data between recruitment phases (Facebook advertising, NDSS email, and HCP email). Where questions were providing little insight and could be improved this was done, and where these changes are relevant, they are noted within the results below.

Table 1 summarises the survey concepts measured and the number of items, per cohort. Questions were asked specifically about the Baby Steps app as it is the only app targeting women with prior GDM that is nationally supported through the healthcare system in Australia.

### 2.4. Data Handling and Analysis

Open-ended responses were managed in Microsoft Excel. Content analysis was used to quantify the presence of concepts in the data (i.e., by generating counts for each code). Initially coding and categorization was conducted by one researcher (AR), with a second researcher reviewing the work (KB). Any discrepancies were discussed and changes made to reflect the agreed categorisation. Discussions with the author team provided a third pass of the analysis.

Data were cleaned and valid survey responses analysed descriptively using the statistical software package IBM SPSS Statistics 28.0.1.0. Summary statistics were calculated (mean ± standard deviation [SD] for normally distributed continuous variables, and frequency [*n*] and percent [%] for categorical variables) separately for the two participant cohorts (i.e., women with GDM and HCPs). In addition, key demographic and clinical characteristics were compared (via t-tests or Chi-square tests) between participants with GDM recruited via NDSS versus paid Facebook advertisements to identify if there were any statistical differences between the two groups.

As this was an exploratory/descriptive study, if participants had missing responses, their data was not excluded if it met the overall valid response criteria outlined above. Valid percent is reported throughout. Chi-square tests were conducted to compare the responses of: (1) the health topics that HCPs and women with prior experience of GDM would want more information on; (2) the preferences of health app users and non-users (though open to health app use) for health app content and functions.

## 3. Results

### 3.1. Response Rates and Sample Characteristics

A total of 1474 eligible, consenting women with GDM completed the survey with valid responses. Facebook paid advertising resulted in 10,222 individual accounts viewing the advert, and 1400 survey link clicks with 916 valid responses (9% translation from advert view to survey completion); while the NDSS direct email resulted in 893 survey link clicks and 558 valid responses (1.3% response rate, not accounting for email open rate). Significant differences between Facebook and NDSS recruited participants were observed, with the latter being older and more culturally diverse (based on reported ethnicity, language spoken at home, and birth country). Table 2 presents the demographic and clinical characteristics for women with prior GDM. Women were predominantly speaking English at home (96%), Australian born (78%), and had experienced GDM in at least one pregnancy (59%).

A total of 179 eligible and 79 valid HCP responses were collected. Table 3 presents the demographic, professional experience, and practice characteristics of HCPs. HCPs were mainly from QLD (70%), female (97%), and with experience of over 10 years (48%).

### 3.2. Health Information Needs and Format following GDM

All participants were asked how they like to receive (women with prior GDM), or provide (HCPs), health information on chronic disease post-pregnancy (Table 4). Most women with prior GDM indicated that they like to receive health information from their doctor (68%) and via email (53%). Apps (28%), including those recommended by a doctor (27%), and information delivered via Facebook groups (21%) were preferred by one in five women, while a minority (8%) indicated that they did not want health information post-pregnancy. Over half of the HCPs support health information delivery via a health app for women with prior GDM.

When asked about referring for health and wellbeing support, HCPs indicated that they most often refer women with prior GDM, to the GP 61% (48), to no one 20% (16), or to free health and wellbeing clinics/programs 13% (10). When asked whether they see any opportunities for improvements to the delivery of health and wellbeing support postpartum (free-text responses), HCP participants most prominently endorsed the need to deliver continued support to women postpartum of GDM (Table 5).

Table 6 presents several health topics that participants (both women with GDM and HCPs) believe would benefit from more detailed information and support postpartum. The top three health topics cited below (healthy eating plans, weight loss/management plans, and prevention of future GDM/T2D) were consistent between HCPs and women with GDM. For all health information topics proposed, however, there was a significant difference between the perspectives of HCPs and women, with a greater percentage of HCPs believing that women would benefit from more information and support across topics. Eleven percent of women (*n* = 266) did not endorse any topic, while there was no HCP that did not endorse at least one topic.

### 3.3. Health App Usage and Preferences

Among participants with prior GDM, 19% (*n* = 273) and 28% (*n* = 400) reported health app usage during pregnancy and post-pregnancy, respectively (total sample *n* = 1474). In total, 33% of the surveyed population (*n* = 492) were health app users. Of the non-health app users (77%), 80% (*n* = 786) reported that they would be open to using a health app recommended by their HCP in the future.

A minority (25%, *n* = 20) of HCPs had recommended apps to women during GDM and post-GDM (14%, *n* = 11), while most HCPs (74%, *n* = 54) indicated that a health app may be useful for women with prior GDM. Among the HCPs who had never recommended a health app to these women (58%, *n* = 46), the majority (73%, *n* = 33) did not know of any reputable apps. Another reason for not recommending these health apps (in free-text responses) was that those apps available were not perceived as meeting women’s needs. Reasons given for this were that they were not culturally relevant, not affordable, or that internet access was limited. In addition to a lack of familiarity with apps, HCPs also suggested recommending apps was not their role or claimed there was limited benefit in using apps. HCPs also reported various barriers they felt women would have in using apps, such as women needing additional support and time, or that some women do not like apps.

### 3.4. Experiences with Baby Steps, the App Nationally Promoted through the Healthcare System for Women with Prior GDM

A small proportion of women with prior GDM (15%, *n* = 220) and HCPs (17%, *n* = 13) had heard of the Baby Steps app, the only nationally promoted digital app for women with prior GDM. Of the women and HCPs that had heard of Baby Steps, 49% (*n* = 108) had tried it and 50% (*n* = 6) had recommended it, respectively. For both HCPs and women, the most common avenue for hearing about Baby Steps was via the NDSS. Women, meanwhile, most commonly heard about Baby Steps from their regular doctor (see Table 7 below).

Of the women that had tried the Baby Steps app, 58% (*n* = 63) indicated the app was useful and were still using it, while 14% (*n* = 15) did not find the app useful. Women were asked to provide feedback on Baby Steps (free-text response) and two main response themes were identified. The first theme related to technical problems surrounding the inability of their smart devices to sync and connect with Baby Steps. The second was the timing of the app, namely when it reached the woman. Respondents reported that if they were “busy with [their] new baby” or they “could not exercise yet”, that hindered their uptake of the Baby Steps app.

Table 8 below presents the preferred health app content and functions among women with prior GDM and HCPs (*n* = 1216). App preferences of women with prior GDM were examined separately from those who reported health app use (either during or after pregnancy) (38%, 462) versus those not using apps but who reported being open to future use of a health app (61%, 754). Health app users indicated that tracking diet, exercise, and weight were the most helpful features in the apps they used (endorsed by ≥42%; see Table 9). The most frequently endorsed preferred app features among non-users (those open to using health apps) were credible health information, suggested exercise routines, and dietary information (endorsed by ≥41%). The proportion of participants endorsing each app feature significantly differed between health app users and non-users (except for leader board competitions).

The most important health app content and functions as reported by HCPs (who believe a health app would be useful for women with prior GDM) (*n* = 54) included reminders to screen for diabetes risk followed by culturally specific information on diet, credible health information, and dietary advice.

Women with prior GDM that were health app users described potential improvements to existing health apps (free text) (Table 9). The most common themes were the need to reduce the cost of current health apps and the inclusion of glucose tracking (with reminders and HCP sharing).

## 4. Discussion

This study describes the use of, and preferences for, health apps among women with prior GDM and HCPs, highlighting content feature and function preferences as well as the role of the HCPs in engagement with a health app. Women with prior GDM want health information provided by their doctor, including recommendations of health apps. While not commonly part of HCP current practice, most HCP participants were open to recommending apps for women post-GDM. There is an overall interest in the use of health apps to provide and receive health information, as well as support for women postpartum of GDM, which was highlighted by HCPs as a current gap in clinical preventative care.

This study contributes to the growing body of research [44,45] which demonstrates that women increasingly desire—and that HCPs also believe women would benefit from—more information to support obesity prevention/treatment (for the prevention of or delayed progress towards T2D) postpartum of GDM. When surveyed, both HCPs and women with GDM expressed healthy eating plans, weight loss/management plans, and advice on the prevention of future GDM/T2D as their top three information topics. However, the study also demonstrates the discrepancy in priorities between HCPs and women with prior GDM. While 75% of HCPs indicated a preference for information on type 2 diabetes risk, only 30% of women expressed a desire for this information postpartum. Qualitative research undertaken with women with prior GDM indicates that women may not have a good understanding of their increased risk of T2D, due to insufficient information and mixed messaging postpartum [46,47]. Women’s low risk perception [48] may also relate to the suboptimal rates of attendance at diabetes screenings 6–12 weeks postpartum [47,49,50], potentially leading to delayed diagnosis and treatment of T2D. Women are often provided with postpartum health behaviour advice during pregnancy, at a time when they may already be overwhelmed with health information [51,52]. To support changes in health behaviour and prevent the onset of chronic disease, women with prior GDM need more information and support postpartum.

In this study, women with prior GDM indicated that HCPs were the preferred source of chronic disease risk reduction advice postpartum of GDM. Another Australian study found that when the information needs of women post-GDM were met by clinicians (e.g., why follow-up screening was necessary), their experiences were generally described more positively, and they were more likely to undertake postpartum diabetes screening [48]. However, research conducted in Australia on the provision of care to women with prior GDM indicates that it is not clear whose role it is to provide postpartum follow-up advice to women with prior GDM [51]. As a result of this lack of clarity, advice provision for women with prior GDM has often been haphazard in nature [51], and similar findings have been reported internationally [53]. Although Australian GPs recognize that they are best suited to provide health advice to women postpartum of GDM [51], the current health system communication pathways have been partly blamed for the gaps in care. HCP participants in the study similarly report that improved communication pathways between HCPs and women with prior GDM are required. However, the HCP participant group included few GPs, allowing for limited primary-case based insights.

In this study, health apps were investigated as an avenue for sharing health advice for physical activity and dietary change among women with prior GDM. Although health apps are used by only a third of women and recommended by only a quarter of HCPs, many others were open to health app use/recommendation. Given that 80% of women not using health apps are open to using a health app that is recommended by their HCP, there is great potential for expanding health app use. By contrast, only 50% of HCPs reported that apps could be an avenue to share health advice post-GDM. This aligns with an Australian study on the use of fitness apps by women, which showed women would be happy to use online health tools if they could be sure they were accurate and backed by medical expertise [34]. HCPs are seen as highly trustworthy sources of information [54], including health app recommendations. However, most health apps are developed and implemented outside of the health system, without the input of HCPs [55]. A study identifying 28,905 weight loss apps found that only 0.05% (17) of the apps had HCP input [55].

The importance of HCP input, and co-creation of digital health resources with users more generally, cannot be understated. There was considerable variation among users when surveyed about preferred app content and features. For instance, and compared to other features, non-app users suggested ‘Credible health information’ as most (45%) desirable, compared with only 18% of current app users. Whereas current app users were most interested in tracking features (such as diet, exercise, and weight). This discrepancy is potentially reflective of the desire for HCP recommendation before using an app. Prior research about app adoption suggests that socio-demographic factors are also correlated with app use [56]. Koivuniemi et al. [56] found that compared with occasional/non-users of a maternal health app, frequent app users were more likely to have a higher education level, better diet quality, be underweight/normal weight, non-smokers, married, and only have one child. More research is thus needed to understand, in addition to HCP recommendation, how postpartum digital health can reach and be relevant across disparate population groups.

As observed in this study and elsewhere [35,57,58], a major barrier to recommending health apps is the lack of knowledge by HCPs of what apps are evidence-based and effective. A UK study identified that those health apps which have been screened, approved, and included as a resource within the health system are preferred by HCPs [57]. To gain this “stamp of approval”, work is being done to develop frameworks for the evaluation of health apps and their positioning within the health system [59]. In Australia, the digital health agency has outlined as one of their strategic priorities the development of a workforce that confidently uses digital health technologies in the delivery of health care by 2025. The emphasis on improving HCPs interaction with apps supports the inclusion of health apps within the health system.

Australians with prior GDM have free access to Baby Steps, which is facilitated and recommended by the NDSS. However, the majority of HCPs in the current study indicated a lack of knowledge of any trustworthy health apps for women post-GDM, and only a minority of both participant groups reported having knowledge of Baby Steps specifically. Thus, current findings suggest limited implementation of Baby Steps, despite it being nationally available. There is need for more research exploring how best to implement mHealth among women with prior GDM, including via health system pathways. The implementation of Baby Steps within Australia without face-to-face support or HCP interaction also requires evaluation. Baby Steps was developed in the UK, where it is accompanied by a structured group program, while the app in Australia is provided on a standalone basis. A randomized control trial (RCT) of the UK program emphasized the importance of peer support to avoid frustration with the app and the importance of a support system [32]. Given the Australian implementation of an app-only approach, and the lack of face-to-face support sessions aimed at building a peer support network, the UK RCT is not generalizable in this context.

There are several study limitations to note. First, the surveys employed non-validated study-specific scales which, although developed by a multidisciplinary team and piloted among the intended population, may not have been valid and reliable assessment tools. However, this approach was appropriate given the lack of pre-existing relevant questionnaires, and the study’s exploratory aims. Furthermore, the free-text questions provided rich accounts of the barriers and incentives for app use among these cohorts. Second, this was a cross-sectional survey completed by a self-selected sample of women with prior GDM (within five years postpartum) and HCPs. Therefore, study results may not reflect the needs/preferences of the broader population of women with prior GDM and/or HCPs, including those from diverse backgrounds. Further, data has not been examined or compared by subgroup (i.e., time since diagnosis; multiple GDM experiences; age; ethnic background; HCP profession; recruitment method). However, the large sample size of women with GDM, and the use of multiple recruitment methods (resulting in a heterogenous sample) is a strength of the study. Third, the HCP survey was completed by a comparatively small cohort, with limited representation of GPs. Thus, further research is needed to examine the generalizability of study findings across health setting, including primary care settings. This smaller sample size also limits the ability to look at the influence of HCP characteristics on responses, limiting further insights. Moreover, there was a high drop-off rate among HCP participants. This may be due to the questions about what apps they recommend to women with prior GDM. Many HCPs do not recommend health apps and therefore such questions may have led to HCPs not believing the survey was relevant for them. HCPs are also time poor, and the survey’s length may have been prohibitive. It is additionally worth noting that some of the women in this study would have experienced GDM and postpartum care during COVID-19, which might have influenced their experiences and left certain of their needs unmet.

## 5. Conclusions

There is an interest from HCPs and women with prior GDM for more health information (including on physical activity and nutrition behaviours) and greater support in obtaining and maintaining a healthy weight post-GDM, thereby preventing future chronic disease. Women are open to engaging with this information in an app, even those women who are currently not using one, particularly when endorsed by HCPs. The majority of women with past experience of GDM want health information provided by their HCP, therefore inclusion of an app within a healthcare system may be an appropriate avenue for dispensing health advice.

## Figures and Tables

**Table 1 nutrients-15-03304-t001:** Concepts, measures, and variables included in the survey for women with prior GDM and HCPs.

Concept	Measure or Variable	Survey Version
*Part 1: Health aims*		
Health goals and achievement	7 items—MC Based on [38]	W
Elaborate on health aim	1 free-text	W
*Part 1: Diabetes preventative care*		
Preventative care provision beliefs	1 item—MC 1 free-text	HCP
*Part 2: Usage of apps*		
Usage/recommendation of health apps	2 items—MC	W, HCPs
Name health apps used/recommended	1 free-text	W, HCPs
Explain usage/recommendation of apps	3 items—MC 1 free-text	W, HCPs
Content and functions	2 items—MC During and post pregnancy Based on [39,40]	W, HCPs
Motivation to use an app	2 items—MC Inspired by HBM	W, HCPs
Baby Steps App	4 items—MC 1 free-text	W
*Part 3: Health system*		
Risk factors	1 MC Based on [4,6]	W
Diagnosis of GDM	2 items—MC	W
Care provider and practice	2 items—MC (W) 5 items—MC (HCP)	W, HCPs
Management of GDM	7 items—MC 1 free-text (W) 5 free-text (HCP)	W, HCPs
Education provided	3 items—MC During and post pregnancy. Based on [6,41] 1 free-text (W) 2 free text (HCP)	W, HCPs
Follow-up	3 item—MC 1 free-text	W, HCPs
Overarching experience	1–3 free-text Positive, negative, and anything else	W, HCPs
*Part 4: About you*		
Demographics	3 free-text (W) 2 items—MC (W) 5-items—MC (HCP) (age, postcode, ethnicity, etc.) Inspired by HBM [42] SES determined by postcode & IRSAD [43]	W, HCPs
GDM experienced/worked in	2 items—MC (W) 5 items—MC (HCP)	W, HCPs
Health-rating	1 item—L	W

HBM Health Belief Model; W women with prior GDM; HCP healthcare professionals; MC multiple choice; L Likert scale; IRSAD Index of relative socio-economic advantage and disadvantage.

**Table 2 nutrients-15-03304-t002:** Demographic characteristics of participants with prior GDM.

Variable	Valid Data	Mean ± SD or % (*n*)
Mean age (SD)	1420	35.6 ±4.9
English spoken at home	1426	95% (1358)
Australian born	1427	77% (1100)
Ethnicity (self-identified)	1252	
Australian		42% (529)
Caucasian		29% (363)
European		12% (145)
Asian		12% (146)
Indigenous/Aboriginal/Torres Strait Islander		2% (25)
Other		4% (44)
State or Territory	1299	
VIC		24% (307)
NSW		24% (313)
QLD		21% (276)
SA/ACT/WA/TAS/NT		31% (403)
Low SES area *	1298	38% (496)
GDM experience	1474	
1st		58% (857)
2nd		35% (515)
3+		7% (102)

* SES area determined by postcode and IRSAD split into low (1–5) and high (6–10) [43].

**Table 3 nutrients-15-03304-t003:** Demographic and GDM experience data of healthcare professional research participants.

Variable	Valid Data	Mean ± SD % (*n*)
Age	58	50.0 years ± 11.3
Female	73	96% (70)
Australian born	76	83% (63)
State or Territory	76	
VIC		8% (6)
NSW		5% (4)
QLD		70% (53)
SA/ACT/WA/TAS/NT		17% (13)
Work location	75	
Metro		35% (26)
Regional		47% (35)
Remote		15% (11)
Other		4% (3)
Type of practice	76	
Private hospital		3% (2)
Public hospital		72% (55)
Private clinic outside hospital		8% (6)
Community clinic		12% (9)
Other		5% (4)
Position	79	
GP		4% (3)
Dietitian		18% (14)
Diabetes Educator		47% (37)
Endocrinologist		6% (5)
Midwife		23% (18)
Nurse		10% (8)
Obstetrician		8% (6)
Management		3% (2)
Other		5% (4)
Time working in GDM	73	
<1 year		3% (2)
1–3 years		10% (7)
3–5 years		16% (12)
5–10 years		23% (17)
10+ years		48% (35)
Currently working in GDM	77	92% (71)
See women with GDM at least weekly	70	84% (59)

**Table 4 nutrients-15-03304-t004:** Desired delivery format of health information following gestational diabetes *.

Preferred Way to Receive/Provide Health Information	Women with Prior GDM (*n* 1474)	HCPs (*n* 79)
Doctor/HCP	68% (1003)	43% (34)
Email ^	53% (294)	NA
Apps	28% (381)	53% (42)
A doctor recommended app ^	27% (156)	NA
Facebook Group ^	21% (107)	NA
Group sessions: in person	11% (167)	NA
Group sessions: virtually	10% (148)	NA
Do not want information	8% (109)	NA
Paper-based handout	NA	38% (30)
Website	NA	41% (32)

* valid percentage reported; ^ response from participants recruited via NDSS (*n* = 558). NA (not applicable) indicates the participant group was not asked this question. HCP indicates healthcare professional.

**Table 5 nutrients-15-03304-t005:** Response themes for improving the delivery of health and wellbeing support postpartum of GDM, as identified by HCPs.

Response Themes:	Count
Improve continuity of support for women after GDM	23
Follow-up to preferably be conducted by the GDM team	6
Reduced cost for women	5
Follow-up incorporated in existing postpartum services (i.e., baby community support, midwifery visits, and playgroups)	3
Easier access to allied health practitioners, including dietitians	3
Increased education for GPs about GDM postpartum care	3
Apps are useful and they can provide connection to the GDM postpartum team	2
Consideration of women living remotely	2

**Table 6 nutrients-15-03304-t006:** Health topics participants want more information on for women following GDM.

Health Information Topics	Women with Prior GDM (*n* 1473)	HCPs (*n* 79)
Weight loss/management plan	41% (597)	65% (51) *
Prevention of gestational diabetes for the next pregnancy	40% (599)	NA
Healthy eating plans	38% (543)	71% (56) *
Social connection and time for self	35% (490)	60% (47) *
Physical activity plans	34% (474)	63% (50) *
Risk of type 2 diabetes	30% (435)	75% (59) *
Sleeping plans	25% (350)	43% (34) *
Breastfeeding	19% (264)	54% (43) *
Glucose tolerance test	17% (250)	56% (44) *

* *p*-value < 0.001. NA (not applicable) indicates the participant group was not asked this question.

**Table 7 nutrients-15-03304-t007:** The avenues through which women and HCPs have heard about Baby Steps.

Avenues	Women with Prior GDM (*n* 220)	HCPs (*n* 13)
National Diabetes Services Scheme	31% (68)	5 (39%)
Regular doctor	30% (66)	NA
Gestational diabetes care team	17% (37)	3 (23%)
Family/friend	12% (27)	NA
Search on the internet	10% (22)	3 (23%)
A client with GDM	NA	0
Other	0	5 (39%)

NA (not applicable) indicates the participant group was not asked this question.

**Table 8 nutrients-15-03304-t008:** Preferred health app content and functions of women with prior GDM, split by health app usage, and HCPs ^†^.

Health App Content and Function	Women with Prior GDM		HCPs # (54)
	Users of Health Apps (*n* 462)	Non-Users of Health Apps ^ (*n* 754)	
Tracking diet	50%	36% *	82% (44)
Tracking exercise	49%	30% *	70% (38)
Tracking weight	42%	33% *	59% (32)
Graphs of tracked information	33%	26% *	61% (33)
Bluetooth/syncing devices	30%	17% *	NA
Suggested exercise routines	25%	45% *	69% (37)
Diet advice	23%	41% *	87% (47)
Credible health information	18%	45% *	87% (47)
Help setting realistic goals	17%	37% *	67% (36)
Coping strategies to deal with daily life	13%	30% *	NA
Reminders to screen for diabetes risk	10%	3% *	93% (50)
Peer support through forums	9%	16% *	65% (35)
Ideas to meet parenting demands	6%	31% *	67% (36)
Leader boards for competition	4%	4%	19% (10)
Others shared GDM experience	4%	14% *	70% (38)
Culturally specific information on diet	NA	NA	87% (47)

* *p*-value < 0.05 for X^2^ of users of health apps versus non-users of health apps. ^ Open to health app use. ^†^ Participant included if response provided for this question. # HCPs who reported a health app would be useful for women with prior GDM.

**Table 9 nutrients-15-03304-t009:** Categories of recommendations for improvements to health apps used by women with (and following) GDM.

Category	Count
Reducing the cost of health apps	14
Inclusion of a glucose level tracker, reminders, and summaries for healthcare team	14
Easier food tracking (e.g., product information up-to-date, easier to input)	11
More health information and new information	10
Better syncing—speed and compatibility	7
Settings for breastfeeding and pregnancy (possibly also GDM)	6
Responsive network (e.g., coach, active community forums)	5
Inclusion of step and dietary tracking	5

## Data Availability

The data presented in this study are available on request from the corresponding author. The data are not publicly available due to ethical obligation and commercial potential of the related work.

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
