# Peer review of "Use and Preferences of Health Apps among Women and Healthcare Professionals Regarding GDM Postpartum Care Related to Diet, Physical Activity, and Weight Management: A Cross-Sectional Survey"

_nutrients, 2023, doi:10.3390/nu15153304_

Round 1
Reviewer 1 Report
To the Authors
I am content to be a reviewer of the original article entitled “Use and preferences of health apps among women and healthcare professionals regarding GDM postpartum care: a cross-sectional survey” which y aims to understand mHealth app use and preferences among women with past GDM and healthcare professionals (HCP) in Australia.
Strengths
- The abstract is well organized and provide solid information about the content of the article
- Offer a clear explanation of the originality of the research, pointing that “evidence on the adoption, acceptability, and effectiveness of such online mHealth tools for women post GDM is limited. Further, what mHealth tools women with prior GDM are currently accessing to support health behavior change is not known, nor is what content and features such women want.” (lines 71-75)
- Objectives and hypothesis of the research are presented in a clear manner (lines 86-89)
- Inclusion and exclusion criteria are well described (lines 82-94)
- Methodology and measured variables used in the study clear described and are adequate to the purpose of the intervention
- Results are clearly presented in a succinct manner
Minor revisions
- Line 112 – misspelled 25 AUD$25
- Methodology of selection of the participants – provide a flow-chart to describe the number of patients invited, the number of mothers excluded or who did not accept participation, the number of patients enrolled, number of persons who do not complete the study, final number of participations.
- For qualitative analysis please specify the statistical package used;
- Describe the content of Baby Steps used by the National Services Schene (NDSS)
Sincerely yours,
Author Response
Minor revisions suggested:
- Line 112 – misspelled 25 AUD$25
Thank you for picking this error up. We have now amended line 115, $25AUD.
- Methodology of selection of the participants – provide a flow-chart to describe the number of patients invited, the number of mothers excluded or who did not accept participation, the number of patients enrolled, number of persons who do not complete the study, final number of participations.
Thank you for the suggestion. Given this was an online survey, we feel that the flow chart approach is not the best use of space for the article. The survey distribution and adoption is described in the Methods and Results as follows. [The results in particular highlighting the number of accounts viewing the survey, the survey clicks and the number of valid responses.]
Line 102 under Methods:
Women with prior GDM were recruited (November 2022 – March 2023) via two main avenues. First, the survey was publicly shared via online social media (Facebook, LinkedIn, Twitter), websites and e-newsletter via the researchers and affiliated organizations, including paid Facebook advertising targeting women aged 18-45. Second, a direct email invitation was distributed by the NDSS to a random sample of 40,000 registrants with prior GDM who had consented to receive information about research opportunities… Participants were only eligible to take part in the survey if 1) they had experienced GDM within the last 5 years and given birth for that pregnancy, 2) had received GDM care within Australia, 3) had not been diagnosed with type 1 diabetes or T2D prior to their pregnancy with GDM and 4) were aged 18 years or older.
Line 187 under Results
A total of 1474 eligible, consenting women with GDM completed the survey with valid responses. Facebook paid advertising resulted in 10,222 individual accounts viewing the advert, and 1400 survey link clicks with 916 valid responses (9% translation from advert view to survey completion); while the NDSS direct email resulted in 893 survey link clicks and 558 valid responses (1.3% response rate, not accounting for email open rate).
- For qualitative analysis please specify the statistical package used;
Thank you for your suggestion. Open-ended responses were managed in Microsoft Excel and the software and procedure described in section 2.3 under Data Handling and Analysis, Line 165.
- Describe the content of Baby Steps used by the National Services Schene (NDSS)
Thank you for your suggestion. On line 72, we have now provided a brief description of the features included in the Baby Steps program.
Author Response
Major Comments:
- Please mention the importance of this study clearly and briefly in the ‘Introduction’.
Line 94 we have included a statement that summarises the importance of the study.
- As the subjects of the study have passed through COVID-19 period, it is very much plausible that COVID-19 might have impacted on GDM and other health issues of the subjects. Did the authors consider this important issue? Do the authors have any data regarding COVID-19 of the entire subjects of the study?
Good point – we have mentioned in the discussion, limitations section, that some women will have experienced GDM and post-partum care during COVID-19 and which might have influenced these experiences and unmet needs (Line 421).
- How reliable was the mHealth app?
Please note that this study was to explore postpartum health information and support needs, along with current use of and preferences for health apps, among women with prior GDM, therefore we do not investigate the reliability of a mHealth app.
- Was the mHealth app able to analyse individual variations of health conditions to provide personalized recommendations effectively?
Please note that this study was to explore postpartum health information and support needs, along with current use of and preferences for health apps, among women with prior GDM, therefore we do not investigate a mHealth app and its ability to analyse individual variations
- Was there any positive impact of using the mHealth app among the subjects compared to the non-user?
Please note that this study was to explore postpartum health information and support needs, along with current use of and preferences for health apps, among women with prior GDM, therefore we do not investigate a mHealth app and compare users and non-users
- Do the authors have any data regarding particular types of diabetes among GDM subjects in their study?
Please note that this study was for women who have experienced gestational diabetes. The inclusion criteria can be found in section 2.1 Study participants on line 116.
Participants were only eligible to take part in the survey if 1) they had experienced GDM within the last 5 years and given birth for that pregnancy, 2) had received GDM care within Australia, 3) had not been diagnosed with type 1 diabetes or T2D prior to their pregnancy with GDM and 4) were aged 18 years or older.
Minor Comment:
- The authors should check their manuscript thoroughly for grammatical errors and typos.
We have reviewed the paper and made any adjustments required.